# LPS-Induced Mortality in Zebrafish: Preliminary Characterisation of Common Fish Pathogens

**DOI:** 10.3390/microorganisms11092205

**Published:** 2023-08-31

**Authors:** Rafaela A. Santos, Cláudia Cardoso, Neide Pedrosa, Gabriela Gonçalves, Jorge Matinha-Cardoso, Filipe Coutinho, António P. Carvalho, Paula Tamagnini, Aires Oliva-Teles, Paulo Oliveira, Cláudia R. Serra

**Affiliations:** 1CIIMAR—Centro Interdisciplinar de Investigação Marinha e Ambiental, Terminal de Cruzeiros do Porto de Leixões, Av. General Norton de Matos s/n, 4450-208 Matosinhos, Portugal; 2FCUP—Departamento de Biologia, Faculdade de Ciências, Universidade do Porto, Rua do Campo Alegre s/n, Ed. FC4, 4169-007 Porto, Portugal; 3i3S—Instituto de Investigação e Inovação em Saúde, Universidade do Porto, R. Alfredo Allen, 208, 4200-135 Porto, Portugal; 4IBMC—Instituto de Biologia Molecular e Celular, Universidade do Porto, R. Alfredo Allen, 208, 4200-135 Porto, Portugal; 5ICBAS—Instituto de Ciências Biomédicas Abel Salazar, Universidade do Porto, R. de Jorge Viterbo Ferreira, 228, 4050-313 Porto, Portugal

**Keywords:** aquaculture, bacterial diseases, lipopolysaccharides (LPS), *Aeromonas hydrophila*, *Photobacterium damselae*, *Tenacibaculum maritimum*, *Vibrio harveyi*

## Abstract

Disease outbreaks are a common problem in aquaculture, with serious economic consequences to the sector. Some of the most important bacterial diseases affecting aquaculture are caused by Gram-negative bacteria including *Vibrio* spp. (vibriosis), *Photobacterium damselae* (photobacteriosis), *Aeromonas* spp. (furunculosis; haemorrhagic septicaemia) or *Tenacibaculum maritimum* (tenacibaculosis). Lipopolysaccharides (LPS) are important components of the outer membrane of Gram-negative bacteria and have been linked to strong immunogenic responses in terrestrial vertebrates, playing a role in disease development. To evaluate LPS effects in fish, we used a hot-phenol procedure to extract LPS from common fish pathogens. *A. hydrophila*, *V. harveyi*, *T. maritimum* and *P. damselae* purified LPS were tested at different concentrations (50, 100, 250 and 500 µg mL^−1^) at 3 days post-fertilisation (dpf) *Danio rerio* larvae, for 5 days. While *P. damselae* LPS did not cause any mortality under all concentrations tested, *A. hydrophila* LPS induced 15.5% and *V. harveyi* LPS induced 58.3% of zebrafish larvae mortality at 500 µg mL^−1^. LPS from *T. maritimum* was revealed to be the deadliest, with a zebrafish larvae mortality percentage of 80.6%. Analysis of LPS separated by gel electrophoresis revealed differences in the overall LPS structure between the bacterial species analysed that might be the basis for the different mortalities observed.

## 1. Introduction

Aquaculture provides a controlled environment to produce aquatic animals and plants, aiming to answer the global human nutritional needs [1]. The occurrence of disease outbreaks represents a significant constraint to the expansion and development of sustainable aquaculture [1]. Gram-negative bacteria are responsible for some of the most common bacterial diseases affecting aquaculture. Examples include *Aeromonas* spp. (furunculosis; haemorrhagic septicaemia), *Vibrio* spp. (vibriosis), *Photobacterium damselae* (photobacteriosis) or *Tenacibaculum maritimum* (tenacibaculosis) [2,3,4,5]. The lack of extensive knowledge regarding virulence factors and host–pathogen interactions for some of these microorganisms, and the wide range of aquaculture fish species produced worldwide, are important factors contributing to the emergence and spread of bacterial infectious diseases [1].

Lipopolysaccharides (LPS), a major component of the outer membrane of Gram-negative bacteria, are considered a significant virulence factor and are linked to strong immunogenic responses in both mammals and teleost fish [6,7,8,9,10,11,12]. LPS are typically constituted by three components: the lipid domain, Lipid A; an oligosaccharide core (attached to Lipid A); and a repeating hydrophilic distal oligosaccharide, known as the O-antigen [10,12]. Lipid A is the most conserved portion of LPS, although variability between bacterial strains can be observed [10,12]. In mammals, this component triggers innate immune responses in the early stages of the infection through the activation of toll-like receptors (TLR) [10,12]. However, given its non-polar nature, Lipid A is only detected by immune cells when released during cell division or bacterial death upon autolysis or phagocytosis [10]. The O-antigen has the highest variability of the three LPS components [10]. In some bacterial strains, LPS lack the O-antigen portion, being named rough LPS (R-LPS), while complete LPS with the three components are named smooth LPS (S-LPS) [10]. O-antigen variability in terms of length and repeats determines the specific immune responses observed between and within bacterial strains [12]. In fact, several studies suggest that the composition and size of the O-antigen are reliable indicators of bacterial virulence during infection and might explain why different bacterial strains cause different immunological responses in the same host [8,10,13,14].

In mammals, LPS, as other pathogen-associated molecular patterns (PAMPs), are recognised by TLRs (especially by TRL4 that is the primary receptor for LPS detection), which trigger the MyD88-dependent pathway, activating nuclear factor-kB (NF-kB), ultimately resulting in the production of pro-inflammatory cytokines [12,14,15,16]. For a successful LPS recognition, TLR4 forms a complex along with three co-stimulatory molecules: the myeloid differentiation protein 2 (MD2), the LPS binding protein (LBP) and the cluster of differentiation 14 (CD14) [12,14,15]. Fish and amphibians, however, show a higher tolerance to LPS when compared to mammals [12,14,15]. Higher doses, within the µg range, are needed to activate the immune cells in fish, whereas, in mammals, only ng are required [12]. A possible explanation for these observations might be that most fish lack the molecules specifically involved in TLR4-mediated LPS recognition and signalling [12]. Some fish are found to express TLR4; however, in those, TLR4 seems to act as a negative regulator of the transcription factor, nuclear factor-kB (NF-kB), through the MyD88-dependent pathway. Simultaneously, the encoding genes for the co-stimulatory molecules, MD2 and CD14, are not found in the available fish genomes, thus suggesting that, in fish, LPS might not be recognised through TLR4-dependent signalling [10,12,14,15].

Independently of the recognition pathways involved, whose knowledge in teleosts is just emerging, LPS from the most common bacterial pathogens affecting aquaculture have not yet been deeply studied. With the exception of a few research works with *Aeromonas hydrophila* [7,17,18,19,20], no other aquaculture pathogens have been the target of dedicated studies to correlate their LPS structure and potential role in virulence and fish mortality. Thus, to start elucidating this subject, in this study, LPS from common fish pathogens, namely *Vibrio harveyi*, *Tenacibaculum maritimum* and *Photobacterium damselae,* were extracted and tested at different concentrations in zebrafish (*Danio rerio*) larvae. Differences in the overall LPS structure from the various bacterial species analysed might be the basis for the different mortalities observed.

## 2. Materials and Methods

### 2.1. Bacterial Strains and Culture Conditions

Five bacterial fish pathogens were used in this study, namely, *Aeromonas hydrophila* LMG 2844; *Tenacibaculum maritimum* LMG 11612; *Pseudomonas aeruginosa* PAO1 LMG 12228; *Vibrio harveyi*, a fish isolate from Nutrition and Immunobiology (NUTRIMU) collection; and *Photobacterium damselae* subsp. *piscicida* strain Lg_h41/01_ [21]. Strains were grown aerobically at 25 °C in tryptic soy broth (TSB, BD Difco) medium, except for *T. maritimum*, which was grown in marine medium (MA, BD Difco).

### 2.2. LPS Extraction and Quantification

Bacterial LPS were extracted using a standard hot phenol–water method [22] with minor modifications. In brief, an inoculum of each bacterial strain (OD_600_ = 0.1) was prepared in 25 mL of TSB or MA medium from overnight cultures. *A. hydrophila, V. harveyi* and *P. damselae* subsp. *piscicida* were grown for 24 h, while *T. maritimum* was grown for 48 h, at 25 °C and 120 rpm. Bacterial pellets were obtained after centrifugation at 18,300 *g* for 10 min at 4 °C and washed in 10 mL of Wash Buffer (205 mM NaCl, 40 mM KCl, 150 mM Na_2_HPO_4_, 20 mM KH_2_PO_4_, 0.15 mM CaCl_2_, 0.5 mM MgCl_2_), followed by centrifugation at 18,300 *g* for 10 min at 4 °C. After two washes, bacterial pellets were dissolved in 10 mL of PBS (137 mM NaCl, 3 mM KCl, 10 mM Na_2_HPO_4_, 1 mM KH_2_PO_4_) and sonicated for 10 min on ice. Then, bacterial cells were subjected to a Proteinase K treatment (100 µg mL^−1^), for 1 h at 65 °C, followed by the addition of RNAse (Sigma-Aldrich, Darmstadt, Germany), 1 µL mL^−1^ of DNase (Roche, Basel, Switzerland) and 4 µL mL^−1^ of 20% (*v*/*v*) MgSO_4_ (Sigma-Aldrich). After overnight incubation at 37 °C, an equal volume of 90% hot phenol (70 ± 2 °C, Sigma-Aldrich) was added to the mix, followed by incubation for 30 min at 70 °C with vigorous shaking. Mixtures were then cooled on ice and centrifuged at 10,400 *g* for 15 min at 4 °C, and the aqueous phase was collected. The remaining LPS in the phenol phase was re-extracted by adding and mixing hot distilled water (70 ± 2 °C), followed by new centrifugation (10,400 *g*, 15 min, 4 °C) and collection of the new aqueous phase. After mixing the two aqueous phases, 0.5 M sodium acetate (Merck, Darmstadt, Germany) was added, followed by an equal volume of absolute ethanol to precipitate the LPS. After the precipitation step (overnight at −20 °C), the suspension was centrifuged at 3000 *g* for 30 min at 4 °C, and the pellet was suspended in 1 mL of autoclaved distilled water. The LPS suspension was then dialysed against distilled water using a 3.5 kDa cut-off membrane (Thermo Fisher Scientific, Waltham, MS, USA) for 24 h at 4 °C. The dialysed suspension was frozen at −80 °C and then lyophilised. The LPS products were weighted and dissolved in PBS to a final concentration of 10 mg mL^−1^. LPS quantification and protein profile was performed as described previously [23]. Briefly, LPS suspensions were 10-fold diluted, resolved on 16% (*w*/*v*) SDS-polyacrylamide gel and stained with the Pro-Q^®^ Emerald 300 Lipopolysaccharide Gel Stain Kit (Life Technologies, Thermo Fisher Scientific, Waltham, MS, USA). LPS was visualised on a GelDoc^TM^ XR^+^ (Bio-rad, Algés, Portugal) system with UV-light radiation. Further, stained gel images were analysed by measuring the pixel’s intensity in each band/sample using ImageJ software v1.54, and quantification was determined by comparing those values with a calibration curve established with commercial *P. aeruginosa* LPS (Sigma-Aldrich L9143 purified by phenol extraction from *P. aeruginosa* ATCC 27316) used as control.

### 2.3. Ethical Statement

All experiments and handling of zebrafish were conducted following the European directive 2010/63/EU for the care and use of laboratory animals and were approved by the Animal Welfare Committee of the Interdisciplinary Centre of Marine and Environmental Research (ORBEA-CIIMAR-27-2019). The work was carried out in a registered installation (N16091.UDER) and performed by trained scientists with FELASA category B.

### 2.4. Zebrafish Husbandry and Breeding

Wild-type zebrafish were maintained in a thermo-regulated water recirculation system and kept under optimal husbandry conditions (water temperature of 28.0 ± 0.5 °C; oxygen level around 7.8 mg L^−1^; ammonia and nitrite levels around 0 mg L^−1^; natural photoperiod of 14 h light:10 h dark). Adult fish were fed twice a day with TetraMin tropical mix (Aquapex, Orni-ex, Vila Nova de Gaia, Portugal), containing 55% of total crude protein and 7% of lipids. Zebrafish embryos were produced by pair-wise mating of adult zebrafish, collected and incubated at 28 °C with natural photoperiod. Zebrafish embryos were kept in EggWater (0.06 g L^−1^ of Instant Oceans) and were daily cleaned. To prevent contaminations, 0.38 mg L^−1^ of methylene blue (Sigma-Aldrich, St. Louis, MO, USA) was added to the EggWater in the first 24 h. After hatching, larvae were kept under the same conditions, and from 5 days post-fertilisation (dpf), larvae were fed twice a day with zebrafeed (Sparos, Olhão, Portugal) containing 60% of total crude protein and 12% of total lipids. At the end of all trials, the remaining larvae were euthanised with an overdose of tricaine methanesulfonate (MS-222, 300 mg L^−1^).

### 2.5. Challenge Tests with LPS and Live Bacteria on Zebrafish

The virulence effects of bacterial LPS and corresponding live bacteria were evaluated using zebrafish (*Danio rerio*) larvae at 3 days post-fertilisation (dpf) as a model organism.

To assess the LPS effect on zebrafish survival, 12 larvae at 3 dpf were distributed in 6-well plates containing 6 mL of EggWater and exposed for 5 days by immersion to different concentrations of LPS from *A. hydrophila* (50, 100, 250, 500 µg mL^−1^), *V. harveyi* (50, 100, 250, 500 µg mL^−1^), *P. aeruginosa* (50, 100, 250 µg mL^−1^), *P. damselae* and *T. maritimum* (50, 100, 250, 500 µg mL^−1^). Cumulative mortalities were registered for 5 days (120 h), and dead larvae were removed. Larvae exposed to 45 µg mL^−1^ LPS of *P. aeruginosa* (Sigma-Aldrich), known to cause at least 50% mortality, were considered as positive control, and larvae kept in EggWater (without LPS exposure) were considered the negative control. The experiment is composed of three independent and biological replicates.

The evaluation of the effect of live bacteria on zebrafish mortality was done as previously described [24] with minor modifications. Briefly, *A. hydrophila*, *V. harveyi* and *Ph. damselae* were cultured for 24 h in TSB at 25 °C with 140 rpm, pelleted by centrifugation (6000 *g*, 15 min, room temperature) and then diluted to different concentrations in PBS. Before the assay, the non-lethal dose (not causing mortality) and the lowest concentration causing 100% mortality were determined by exposing zebrafish to a wide range of bacterial concentrations (10^5^ up to 10^10^). A total of 10 larvae at 6 dpf were distributed in 6-well plates containing 6 mL of EggWater and exposed by immersion to *A. hydrophila* (1 × 10^10^, 5 × 10^9^, 1 × 10^9^), *V. harveyi* (1 × 10^9^, 5 × 10^8^, 1 × 10^8^) and *Ph. damselae* (5 × 10^7^, 2 × 10^7^, 1 × 10^7^). After 24 h of challenge, larvae were transferred to new and sterile EggWater and cumulative mortalities were registered for 5 days (120 h) with the dead larvae removed. Larvae exposed to PBS and EggWater were considered as control groups. The experiment was composed of three independent and biological replicates.

### 2.6. Statistical Analysis

Survival data were plotted using Kaplan–Meier, and differences between treatments were accessed using the log-rank test and a significant level of 0.05 in the GraphPad Prism v8 software (Stortford, UK). 

## 3. Results

### 3.1. Fish Pathogens A. hydrophila, V. harveyi, T. maritimum and P. damselae subsp. piscicida Have Different Lipopolysaccharides (LPS) Profiles

The LPS profile of *A. hydrophila*, *V. harveyi*, *T. maritimum*, *P. damselae* subsp. *piscicida* and *P. aeruginosa* extracted (hereafter Ext) was evaluated by resolving different quantities (2, 5 and 10 µL) of hot phenol–water-extracted LPS on a 16% (*w*/*v*) SDS-polyacrylamide gel (Figure 1). To assess extractability, commercial LPS from *P. aeruginosa* extracted by phenol methodology was used as control (hereafter referred to as Com). The band profile of the LPS extracted from *P. aeruginosa* (Ext) is similar to the LPS commercially available from *P. aeruginosa* (Com), showing both smooth (S-LPS) and rough LPS (R-LPS). Nonetheless, S-LPS extracted from *P. aeruginosa* showed a lower molecular weight than the S-LPS commercially available from *P. aeruginosa.* On the other hand, when comparing the LPS profile of *A. hydrophila*, *V. harveyi*, *T. maritimum* and *P. damselae* subsp. *piscicida* with the LPS from *P. aeruginosa,* the absence of S-LPS is clear. The R-LPS from *A. hydrophila* and *V. harveyi* have lower molecular weight accumulating in the lower part of the gel, while the *P. damselae* subsp. *piscicida* R-LPS have a high molecular weight. Additionally, in *T. maritimum,* only two LPS bands could be detected.

### 3.2. Virulence of A. hydrophila, V. harveyi, P. damselae subsp. piscicida and T. maritimum Cells and LPS Is Highly Variable and Strain Specific

To assess LPS contribution to fish pathogen virulence and larvae-induced mortality, two different assays were performed in which zebrafish larvae were exposed to (i) different concentrations of live bacteria (following our own previously established infection protocols) for 24 h or (ii) increased concentrations of corresponding LPS (ranging from 50 µg mL^−1^ to 500 µg mL^−1^) for up to 5 days.

In all the experiments, a group composed of larvae kept in EggWater and non-exposed to live bacteria or LPS was used as negative control, having a total survival rate of 99.4%. Additionally, the LPS from *P. aeruginosa* (Com) were used as positive control at 45 µg mL^−1^ since it has been previously described to cause ~50% and ~80% mortality in larvae after 8 h and 24 h of exposure, respectively [25]. In all our experiments, this control showed the highest virulence, causing a cumulative mortality rate of 83% during the 120 h experiment.

Zebrafish larvae infected with 1 × 10^10^ and 5 × 10^9^ CFU mL^−1^ of *A. hydrophila* began to show mortalities after 16 h post-infection (hpi) and progressed through time, reaching ~60% and ~15%, respectively, at 120 hpi (Figure 2). By comparing with the control group after the 120 h of trial, these two concentrations of *A. hydrophila* significantly decrease zebrafish survival (*p* < 0.05; Appendix A). On the contrary, challenge with the lowest concentration of *A. hydrophila* used in this study (1 × 10^9^ CFU mL^−1^) did not induce any mortalities in zebrafish larvae. Accordingly, only the highest concentration of LPS extracted from *A. hydrophila* (500 µg mL^−1^) was able to induce significant mortalities on zebrafish larvae (~15%; *p* = 0.0005) after 96 hpi when compared to the control group composed of non-exposed larvae (Figure 2, Appendix A).

The virulence of *V. harveyi* on zebrafish larvae was evaluated as illustrated in Figure 3. Challenge using *V. harveyi* bacterial cells induced mortalities in zebrafish larvae that progressed rapidly, reaching 100% after only 2 h of exposure to 1 × 10^9^ and 5 × 10^8^ CFU mL^−1^. Moreover, the *V. harveyi* inoculum of 1 × 10^8^ CFU mL^−1^ started to induce mortalities after 16 hpi, reaching ~25% at 120 hpi (Figure 3). When comparing with the control group, composed of non-exposed larvae, these results showed that all the tested concentrations of *V. harveyi* cells induced significant mortalities on zebrafish larvae (*p* < 0.001; Appendix A). LPS extracted from *V. harveyi* was able to induce mortalities on zebrafish larvae when exposed not only to 500 but also to 250 µg mL^−1^ (*p* < 0.01; Appendix A). Zebrafish larvae exposed to 500 µg mL^−1^ of LPS extracted from *V. harveyi* started to show mortalities after 72 hpi and progressed until the end of the trial (120 h) with an average mortality rate of ~58% (Figure 3). Additionally, 250 µg mL^−1^ of LPS extracted from *V. harveyi* induced mortalities (~11.1%) after 120 hpi.

Zebrafish larvae exposed to 5 × 10^7^ and 2 × 10^7^ CFU mL^−1^ of *P. damselae* subsp. *piscicida* started to show mortalities after 16 hpi, progressing through time, and reaching ~78% and ~20%, mortality respectively at 22 hpi (Figure 4). Due to *P. damselae* subsp. *piscicida* Lg_h41/01_ slower growth rate, the extraction of LPS resulted in lower amounts when compared to the other bacterial pathogens. Thus, the effect of *P. damselae* LPS on zebrafish survival was determined by testing only 2 different LPS concentrations (100 and 500 µg mL^−1^). As illustrated in Figure 4B, LPS from *P. damselae* subsp. *piscicida* did not induce significant mortalities in zebrafish larvae (~2.8%) after 120 hpi, independently of the concentration used, when compared to the control group composed of non-exposed larvae (Appendix A). On the contrary, live bacterial exposure resulted in a significant zebrafish mortality (*p* < 0.05, Appendix A).

Since *T. maritimum* is a marine fish pathogen that requires high salt concentrations (>30 g L^−1^) to grow and survive in the environment [26], we first tested *T. maritimum* cells stability in zebrafish EggWater (that contains only 0.06 g L^−1^ of NaCl) for 24 h and discovered them to be unable to survive under our experimental conditions. On the other hand, increasing EggWater salt concentration to the needs of *T. maritimum* was deleterious to zebrafish. Thus, for this pathogen, we only tested the virulence effect of the extracted LPS (Figure 5). The 500 µg mL^−1^ of *T. maritimum* LPS started to induce mortalities after 96 hpi that progressed through the experiment, reaching 80.6% after 120 hpi (*p* < 0.001; Appendix A).

Finally, after observing that the commercial LPS from *P. aeruginosa* (PA-Com) showed, in all our experiments, a much higher virulence than the extracted LPS from fish pathogens, we decided to evaluate if the LPS extracted manually from *P. aeruginosa* (PA Ext) could induce the same toxic effects on zebrafish. As illustrated in Figure 6, larvae exposed to 100 µg mL^−1^ and 50 µg mL^−1^ of commercial LPS showed 100% mortality after 2 and 4 h post-incubation, respectively. On the other hand, when larvae were exposed to 250 µg mL^−1^ and 100 µg mL^−1^ of extracted LPS from *P. aeruginosa*, mortalities only began after 48 hpi and did not surpass 2.8% and 5.6%, respectively. By comparing with the control group, composed of non-exposed larvae, these results showed that the LPS extracted from *P. aeruginosa* did not induce any virulence of zebrafish larvae (Appendix A), whereas the commercial LPS is highly virulent under the tested conditions (*p* < 0.001; Appendix A).

## 4. Discussion

LPS are often considered one of the most important virulence factors in Gram-negative bacteria [10]. LPS contribution to the virulence of aquaculture pathogens is, however, far from elucidated. In this study, immersion trials with zebrafish larvae including both live cells and LPS extracted from problematic fish bacterial pathogenic species, such as *A. hydrophila* (furunculosis), *V. harveyi* (vibriosis), *P. damselae* subsp. *piscicida* (photobacteriosis) and *T. maritimum* (tenacibaculosis), were tested simultaneously.

Very distinguishable LPS profiles were obtained from strains *A. hydrophila* LMG 2844, *V. harveyi* isolate, *P. damselae* subsp. *piscicida* Lg_h41/01_ and *T. maritimum* LGM 11612, producing only R-LPS. To rule out extractability issues, LPS from a *Pseudomonas aeruginosa* strain (PAO1 LMG 12228, ATCC 15692) were also manually extracted and compared to the commercially available LPS from *P. aeruginosa* (Sigma-Aldrich L9143) obtained using a similar extraction methodology (phenol-based) but from a different strain (ATCC 27316). While the R-LPS of *P. aeruginosa* PAO1 LMG 12228 and those commercially available LPS from *P. aeruginosa* ATCC 27316 appeared to be very similar, the S-LPS region revealed observable differences. S-LPS from *P. aeruginosa* LMG 11612 had their bands more dispersed and with lower molecular weight than bands of S-LPS from commercially available *P. aeruginosa*. Because the composition and the size of the O-antigen (part of S-LPS) are considered determinants of bacterial virulence [6,10,13,27], such differences in LPS structure between the two *P. aeruginosa* strains may explain the significant differences observed in the zebrafish larvae survival assay, with 100 µg mL^−1^ of commercial *P. aeruginosa* LPS resulting in 100% mortality within only 2 h, whereas 250 µg mL^−1^ LPS of *P. aeruginosa* PAO1 LMG 11612 led to only 2.8% mortality after 120 h. Nevertheless, by using the *P. aeruginosa* control strain (whose LPS were extracted simultaneously with LPS from other bacterial strains) and obtaining a similar profile to *P. aeruginosa* LPS commercially available, we confirmed that the LPS extraction methodology used in our study is able to extract both S-LPS and R-LPS, thus ruling out extractability issues. 

The LPS profiles obtained for *A. hydrophila* LMG 2844, *V. harveyi* isolate, *P. damselae* subsp. *piscicida* Lg_h41/01_ and *T. maritimum* LGM 11612 under our experimental conditions produced only R-LPS. A similar observation of low molecular-weight LPS has been previously described for *V. harveyi* [28,29] and for *T. maritimum* O3 serotype strains [30], although with a higher number of bands, but not for *T. maritimum* O1 and O2 serotypes, to which the type of strain of *T. maritimum* NCIMB 2154^T^ (LMG 11612) used in our study belongs [31]. The LPS profile of *T. maritimum* O1 and O2 serotypes detected by Western blot analysis revealed a typical laddering pattern, although without a clear separation of S-LPS and R-LPS [32]. LPS detection by Western blot analysis (a much more sensitive technique), rather than by staining, might be the reason for failing to observe a typical LPS pattern in our study. In fact, LPS staining approaches, such as silver stain, have been previously described to work differently depending on the target strain, being successful in detecting LPS from control *E. coli* ATCC 25922 but failing to reveal clear laddering for LPS of *T. maritimum* strains [33]. Thus, we cannot rule out that a similar phenomenon might have happened under our experimental conditions, using the Pro-Q^®^ Emerald 300 Lipopolysaccharide Gel Stain Kit, for the LPS obtained for *A. hydrophila* LMG 2844, *V. harveyi* isolate, *P. damselae* subsp. *piscicida* Lg_h41/01_ and *T. maritimum* LGM 11612 while working perfectly with the LPS extracted from *Pseudomonas aeruginosa* strain (PAO1 LMG 12228, ATCC 15692). The fact that we could not detect high molecular-weight LPS in *A. hydrophila*, as previously described for strains of different serotypes [18,19,20,34], might be an indication of this detection failure. Additionally, the Pro-Q^®^ Emerald 300 Lipopolysaccharide Gel Stain Kit reacts with periodate-oxidised carbohydrate groups, and it is possible that the S-LPS carbohydrates of our strains are not oxidised, thus not being stained. On the other hand, the obtained low molecular-weight R-LPS might also represent the real LPS pattern for the strains under study, since the absence of S-LPS is not an unprecedant observation. As previously said, both *V. harveyi* [28,29] and *T. maritimum* O3 serotype strains [30] present only low molecular-weight R-LPS, and although this is the first analysis of *P. damselae* subsp. *piscicida* LPS, a closely related species named *P. phosphoreum* was also reported to present only low molecular-weight R-LPS [35]. In addition, we observed in a previous work [23] that the extraction method used in this study recovers efficiently the polar forms of LPS (S-LPS) from the aqueous phase, and the extracted LPS profiles obtained for *Pseudomonas aeruginosa* strain PAO1 (LMG 12228), presenting both S- and R-LPS and a similar profile as the commercial LPS, which corroborates this analysis. In mammals, both R-LPS and S-LPS trigger a strong innate immune response, yet by different signalling mechanisms [9,36]. Whether this is also true for fish remains unclear, since the LPS recognition mechanism in fish is noticeably different and not yet fully understood [12,15,36].

The virulence of the different LPS extracted from the target aquaculture pathogens under study was highly variable, strain-specific and not directly correlated with the mortalities observed when exposing zebrafish larvae to live cells of the corresponding strains. *T. maritimum*, *V. harveyi* and *P. damselae* subsp. *piscicida* are all marine fish pathogens, while *A. hydrophila* can be found in both fresh and marine waters. Although zebrafish are a fresh-water fish species, they were also susceptible to the tested marine pathogens and their LPS. Cells of *P. damselae* subsp. *piscicida* Lg_h41/01_ were the deadliest ones, with a 5 × 10^7^ CFU mL^−1^ inoculum being sufficient to induce more than 50% mortality in zebrafish larvae (Figure 7A). To achieve similar levels of mortality with *V. harveyi* or *A. hydrophila*, it was necessary to use 5 × 10^8^ CFU mL^−1^ and 1 × 10^10^ CFU mL^−1^ of bacterial inoculum, respectively (Figure 7A). On the contrary, LPS extracted from *P. damselae* subsp. *piscicida* did not induce significant mortalities, even when using a high concentration of 500 ug mL^−1^ (Figure 7B). A recent study with *P. damselae* subsp. *piscicida* revealed that its extracellular products are highly toxic to red-banded seabream, but such toxicity is independent of LPS [37], thus corroborating our results that the LPS from this species might not be a potent virulent factor in fish.

The aquaculture pathogen with the deadliest LPS to zebrafish larvae was *T. maritimum* (80.6% mortality when using 500 µg mL^−1^ LPS; Figure 7B), and the virulence of the live cells could not be tested due to the susceptibility of *T. maritimum* cells to the low-salt EggWater used for zebrafish larvae survival experiments. This observation suggests that, contrary to *P. damselae* subsp. *piscicida*, LPS is a key virulence factor in *T. maritimum*. Although LPS from different *T. maritimum* strains have been previously characterised [30,31,32], their contribution to *T. maritimum* virulence and disease establishment and fish mortality was for the first time addressed in this study. Further studies, using susceptible marine fish species allowing to simultaneously test *T. maritimum* cells and LPS, will shed light on the contribution of LPS to *T. maritimum* virulence.

Given the 58.3% mortality observed in zebrafish larvae when challenged with 500 µg mL^−1^ LPS of *V. harveyi*, LPS are also likely an important virulent factor in this species. Despite being one of the most problematic aquaculture pathogens [2,38,39,40] and one of the most studied bacterial species, in particular to understand quorum sensing [41,42,43], little is still known about *V. harveyi* LPS and their contribution to virulence and disease in fish. Our work suggests it is worth investigating.

## 5. Conclusions

In summary, our results demonstrate that LPS extracted from different bacterial aquaculture pathogens have very distinguishable structures and virulence potentials. The mortalities observed when exposing zebrafish larvae to purified LPS did not correlate with the virulence observed when exposing zebrafish larvae to the corresponding live bacteria. Under our experimental conditions, LPS from *V. harveyi* and *T. maritimum* are a major virulence factor, whereas LPS from *P. damselae* subsp. *piscicida* are not, although cells of the latter are the deadliest ones for zebrafish larvae. The LPS from *P. damselae* subsp. *piscicida* were for the first time isolated and, together with LPS from *T. maritimum* and *V. harveyi* (previously isolated but not tested for mortality induction), were all tested for the first time in vivo to establish their virulence in fish. New studies are now underway to further clarify our observations.

## Figures and Tables

**Figure 1 microorganisms-11-02205-f001:**
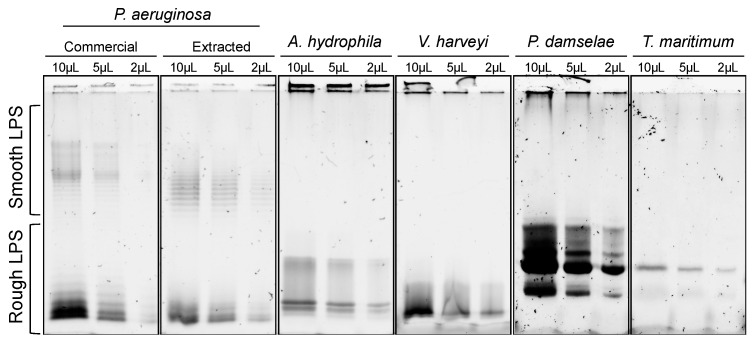
Band profile of the lipopolysaccharides (LPS) extracted from *A. hydrophila, V. harveyi*, *T. maritimum*, *P. damselae* subsp. *piscicida* and *P. aeruginosa* (Extracted). LPS from bacterial cultures were extracted using a standard phenol method, resolved on a 16% (*w*/*v*) SDS-polyacrylamide gel and stained with the Pro-Q^®^ Emerald 300 Lipopolysaccharide Gel Stain Kit. Commercial LPS from *P. aeruginosa* (Commercial) were used as reference. Smooth LPS represent high molecular-weight LPS with a complex structure (composed of the O-antigen, a core polysaccharide chain and Lipid A), while the rough LPS represent low molecular-weight LPS with a simpler structure (composed only of a core polysaccharide chain and Lipid A).

**Figure 2 microorganisms-11-02205-f002:**
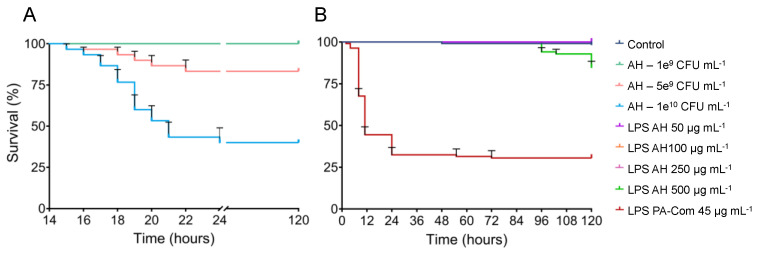
Effects of live bacteria (**A**) or lipopolysaccharides (**B**) of *Aeromonas hydrophila* on zebrafish survival. To evaluate *A. hydrophila* virulence on zebrafish larvae, 6 dpf zebrafish larvae were exposed for 1 day to different concentrations of *A. hydrophila* (1 × 10^9^, 5 × 10^9^, 1 × 10^10^ CFU *A. hydrophila* mL^−1^) by immersion. To assess the LPS effect on zebrafish survival, 3 dpf zebrafish larvae were exposed for 5 days by immersion to different concentrations of extracted LPS from *A. hydrophila* (50, 100, 250, 500 µg mL^−1^). Commercial LPS from *P. aeruginosa* (PA-Com) at 45 µg mL^−1^ were used as a positive control, and unchallenged larvae kept in EggWater were used as a negative control (Control). Cumulative mortalities were registered for 5 days (120 h). Data are composed of three independent biological experiments.

**Figure 3 microorganisms-11-02205-f003:**
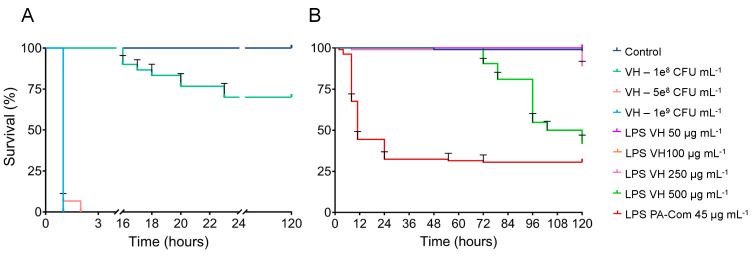
Effects of live bacteria (**A**) or lipopolysaccharides (**B**) of *Vibrio harveyi* on zebrafish survival. To evaluate *V. harveyi* virulence on zebrafish larvae, 6 dpf zebrafish larvae were exposed for 1 day to different concentrations of *V. harveyi* (1 × 10^8^, 5 × 10^8^, 1 × 10^9^ CFU *V. harveyi* mL^−1^) by immersion. To assess the LPS effect on zebrafish survival, 3 dpf zebrafish larvae were exposed for 5 days by immersion to different concentrations of extracted LPS from *V. harveyi* (50, 100, 250, 500 µg mL^−1^). Commercial LPS from *P. aeruginosa* (PA-Com) at 45 µg mL^−1^ was used as positive control, and unchallenged larvae kept in EggWater were used as negative control (Control). Cumulative mortalities were registered for 5 days (120 h). Data is composed of three independent biological experiments.

**Figure 4 microorganisms-11-02205-f004:**
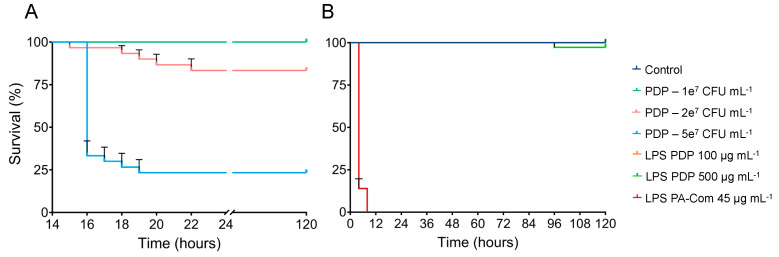
Effects of live bacteria (**A**) or lipopolysaccharides (**B**) of *Ph. damselae* subps. *piscicida* on zebrafish survival. To evaluate *Ph. damselae* subps. *piscicida* virulence on zebrafish larvae, 6 dpf zebrafish larvae were exposed for 1 day to different concentrations of *Ph. damselae* subps. *piscicida* (1 × 10^7^, 2 × 10^7^, 5 × 10^7^ CFU *Ph. damselae* subps. *piscicida* mL^−1^) by immersion. To assess the LPS effect on zebrafish survival, 3 dpf zebrafish larvae were exposed for 5 days by immersion to different concentrations of extracted LPS from *Ph. damselae* subps. *piscicida* (100, 500 µg mL^−1^). Commercial LPS from *P. aeruginosa* (PA-Com) at 45 µg mL^−1^ were used as positive control, and unchallenged larvae kept in EggWater were used as negative control (Control). Cumulative mortalities were registered for 5 days (120 h). Data are composed of three independent biological experiments.

**Figure 5 microorganisms-11-02205-f005:**
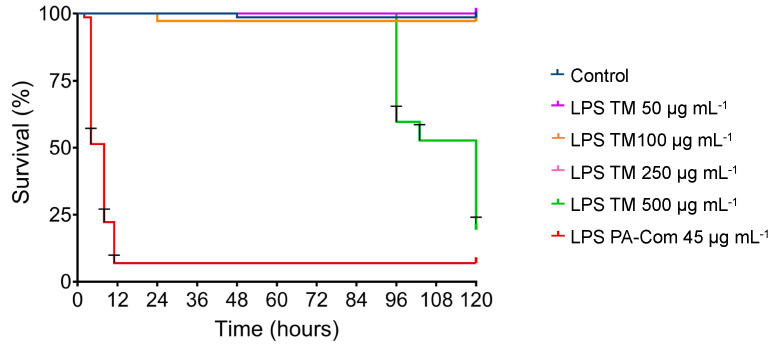
Effects of lipopolysaccharides from *Tenacibaculum maritimum* on zebrafish survival. The 3 dpf zebrafish larvae were exposed for 5 days by immersion to different concentrations of LPS extracted from *T. maritimum* (50, 100, 250, 500 µg mL^−1^). Commercial LPS from *P. aeruginosa* (PA-Com) at 45 µg mL^−1^ were used as positive control, and unchallenged larvae kept in EggWater were used as negative control (Control). Cumulative mortalities were registered for 5 days (120 h). Data are composed of three independent biological experiments.

**Figure 6 microorganisms-11-02205-f006:**
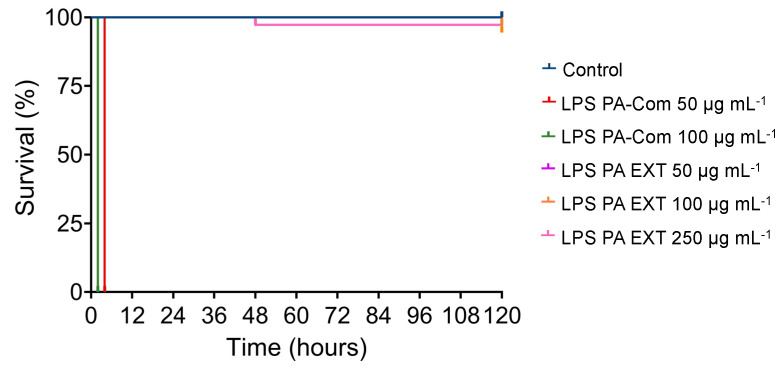
Effects of lipopolysaccharides from *Pseudomonas aeruginosa* on zebrafish survival. The 3 dpf zebrafish larvae were exposed for 5 days by immersion to different concentrations of LPS commercially available from *P. aeruginosa* (LPS PA-Com at 50 and 100 µg mL^−^^1^) and LPS extracted from *P. aeruginosa* (LPS PA-EXT at 50, 100 and 250 µg mL^−^^1^). Unchallenged larvae kept in EggWater were used as negative control (Control). Cumulative mortalities were registered for 5 days (120 h). Data are composed of one independent biological experiment with three technical replicates.

**Figure 7 microorganisms-11-02205-f007:**
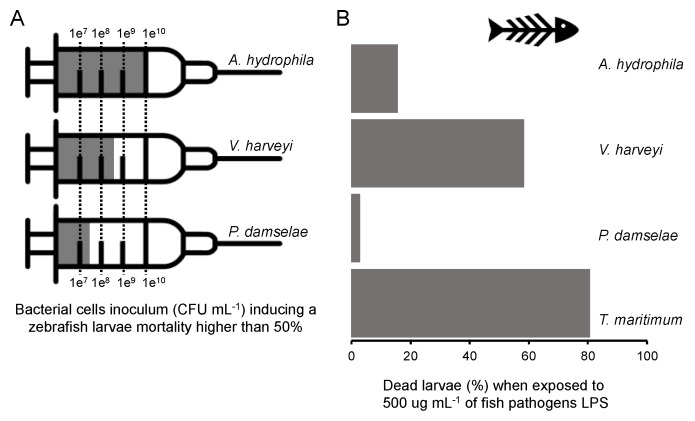
(**A**) Fish pathogen (*A. hydrophyla*, *V. harveyi*, *P. damselae* subps. *piscicida*) bacterial cells inoculum (CFU mL^−1^) used to induce a zebrafish larvae mortality higher than 50%. (**B**) The % of dead larvae obtained after challenging with 500 µg mL^−1^ LPS purified from fish pathogens *A. hydrophyla*, *V. harveyi*, *P. damselae* subps. *piscicida* and *T. maritimum*.

## Data Availability

Relevant data are included in the article and Appendix A.

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
