# Peer review of "LPS-Induced Mortality in Zebrafish: Preliminary Characterisation of Common Fish Pathogens"

_microorganisms, 2023, doi:10.3390/microorganisms11092205_

Round 1

Reviewer 1 Report

In this manuscript, the authors evaluated the virulence of LPS from common fish pathogens (A. hydrophila, V. harveyi, T. maritimum and P. damselae) by challenge tests with LPS. Differences in the overall LPS structure from these bacterial species were also analyzed. The results demonstrate that LPS extracted from different bacterial aquaculture pathogens have very distinguishable structures and virulence potential. However, the manuscript need to be improved before considering it for publication.

1. line 166-167, “different concentrations of LPS from A. hydrophila (50, 100, 250, 500, 750 µg mL-1), V. harveyi 166 (50, 100, 250, 500 µg mL-1),” why not use the concentration gradient?

2. line 165, “containing 6 mL of EggWater and exposed for 5 days by immersion…”, did any EggWater change?

3. line 99-101, “Strains were grown aerobically at 25°C in tryptic soy broth…”, line “A. hydrophila, V. harveyi, Ph. damselae were cultured for 24 h in BHI”, why not use the same medium.

4. line 180, “…exposed for 1 day…” means that the fish was transferred to the EggWater without bacteria after immersion infection for 1 days. Please provide more details.

5. line 183 “Larvae exposed to 1 x PBS…as control groups” should be “Larvae exposed to EggWater…as control groups”.

6. In Figure 1, the tree bands mean the different quantities (2, 5 and 10 µL) of hot phenol-water-extracted LPS? The authors should make it clear in figure 1.

7. Please check the format of References.

Minor editing of English language required.

Author Response

We would also like to express our sincere thanks to the reviewer for the constructive comments to improve our manuscript overall quality. We have studied each comment carefully and made the suggested corrections, accordingly, answering the questions in a point-by-point manner, as listed below.

Q1: line 166-167, “different concentrations of LPS from A. hydrophila (50, 100, 250, 500, 750 µg mL-1), V. harveyi 166 (50, 100, 250, 500 µg mL-1),” why not use the concentration gradient?

A1: The reviewer is completely right. We apologize for the mistake. LPS from fish pathogens were tested using concentration gradients of 50, 100, 250, and 500 µg mL-1. The 750 µg mL-1 is an error. We have corrected this error accordingly.

Q2: line 165, “containing 6 mL of EggWater and exposed for 5 days by immersion…”, did any EggWater change?

A2: Zebrafish experiments followed the guidelines of regular care, and eggwater was changed daily. Thus, we prepared “stock solutions” with different LPS concentrations in EggWater, and we changed the eggwater in each well daily to ensure water quality during all the trials.

Q3: line 99-101, “Strains were grown aerobically at 25°C in tryptic soy broth…”, line “A. hydrophila, V. harveyi, Ph. damselae were cultured for 24 h in BHI”, why not use the same medium.

A3: The reviewer is right. As detailed in Section 2.1., all bacteria were cultivated in TSB, with the exception of T. maritimum, which grew in marine medium. We have corrected this error accordingly.

Q4: line 180, “…exposed for 1 day…” means that the fish was transferred to the EggWater without bacteria after immersion infection for 1 days. Please provide more details.

A4: We agree with the reviewer’s comment. We have added the following explanation to the manuscript on line 182: ”After 24h of challenge, larvae were transferred to new and sterile EggWater and cumulative mortalities were registered for 5 days (120 h) with the dead larvae removed”.

Q5: line 183 “Larvae exposed to 1 x PBS…as control groups” should be “Larvae exposed to EggWater…as control groups”.

A5: We thank the reviewer for the comment. Since the different bacterial concentrations were prepared in 1xPBS, we exposed the bacteria to 1xPBS do discard any effect of PBS. Additionally, we included another control (non-exposed larvae), in which the larvae were kept in EggWater. For clarity, we changed the sentence to “Larvae exposed to 1 × PBS and EggWater”.

Q6: In Figure 1, the tree bands mean the different quantities (2, 5 and 10 µL) of hot phenol-water-extracted LPS? The authors should make it clear in figure 1.

A6: We agree with the Reviewer’s comment. We added that information to the Figure.

Q7: Please check the format of References.

A7: All references were checked, and errors corrected.

Reviewer 2 Report

In this study, authors describe " LPS-induced mortality in zebrafish:  preliminary characterisation of common fish pathogens". Based on this field, the Ms could be revised before being published in this magazine.

1.     Fig. 1, the target band of different species was in the different gels, and there is no marker, the size of protein is different to be known. Markers should be added or the proteins in different species should be in the same gel.

2.     Fig. 1, Plugging phenomenon occurred in A. hydrophila V. harveyiP. damselae, the possible reason should be analyzed.

3.     Significantly different result of LPS from commercial P. aeruginosa in Fig. 4B and Fig.2B /3B? Authors should explain it.

4.     The lethal affected by LPS extracted from marine was more than that from freshwater, while zebrafish belong to freshwater fish. Authors should explain the potential reason in the discussion.

English is OK. 

Author Response

We would also like to express our sincere thanks to the reviewer for the constructive comments to improve our manuscript overall quality. We have studied each comment carefully and made the suggested corrections, accordingly, answering the questions in a point-by-point manner, as listed below.

Q1:  Fig. 1, the target band of different species was in the different gels, and there is no marker, the size of protein is different to be known. Markers should be added or the proteins in different species should be in the same gel. 

A1: One of the main purposes of this study was to analyze the LPS profile of important fish pathogens. In Figure 1, we analyzed the band profile of purified lipopolysaccharides (LPS), and so, no proteins were loaded onto the gels (although isolated LPS are frequently not 100% pure, and protein contaminants can co-purify). Thus, the use of a protein marker was not suitable for this experiment, as the gels show only lipopolysaccharides and not proteins. Nevertheless, because we agree with the reviewer and acknowledge the importance of a marker as control, in every gel we ran we used the commercially available LPS from P. aeruginosa as LPS marker. Although Figure 1 is a composition of several gel-images, for clarity, we send the original images so that the reviewer can assess this.

Q2: Fig. 1, Plugging phenomenon occurred in A. hydrophila V. harveyiP. damselae, the possible reason should be analyzed. 

A2: We apologize, but we do not know exactly what the reviewer means by “plugging phenomenon”.  1) If the reviewer is referring to the lack of a clear band separation in the gel, like a “smear” aspect, we consider that it can be a matter of inefficient separation of biomolecules. For example, if we look at P. damselae gel, we see that the “smear” only appears in the upper bands, but not in the lower one. The electrophoretic separation of biomolecules with sugars is not always as clear-cut as it is for other molecules. Thus, it can be assumed, due to the heterogeneity of the smear distribution in these lanes, that the smear will be related to the existence of different oligosaccharide subunits, which when separated in the polyacrylamide gel cannot be resolved. 2) If the reviewer is referring to the “accumulation” of stained material in the well, this could result of non-LPS components that might have been stained with the kit used. Isolated LPS are frequently not 100% pure and, therefore, can be complexed with something that prevents them from entering the gel. Other molecules, namely proteins, can be co-purified during LPS extraction.

Q3: Significantly different result of LPS from commercial P. aeruginosa in Fig. 4B and Fig.2B /3B? Authors should explain it.

A3: Although zebrafish have many advantages as model organisms, the early life stages of zebrafish are the most sensitive stage of development. Although we followed the same experimental procedures in all animal trials, we observed some variability between experiments, probably due to biological variability. Also, because zebrafish have higher sensitivity in this developmental stage, we cannot rule out some technical problems. We tried in every experiment to minimize this by using three independent biological replicates (larvae from 3 different spanning) and 3 technical replicates within each biological replicate, but still variation could be observed. Similar results have been reported before (see Matinha-Cardoso et al., 2022).

Q4: The lethal affected by LPS extracted from marine was more than that from freshwater, while zebrafish belong to freshwater fish. Authors should explain the potential reason in the discussion. 

A4: T. maritimum, V. harveyi and P. damselae subsp. piscicida are all marine fish pathogens, while only A. hydrophila can be found in both fresh and marine waters. When looking for the lethal effect of LPS purified from fish pathogens, T. maritimum was the deadliest followed by V. harveyi, A. hydrophila and finally P. damselae. Since the LPS from P. damselae subps. piscicida, a marine fish pathogen, is less virulent, and based only on the four tested pathogens, we cannot conclude that zebrafish are more affected by marine than freshwater pathogens. The sentence “…T. maritimum, V. harveyi and P. damselae subsp. piscicida are all marine fish pathogens, while A. hydrophila can be found in both fresh and marine waters. Although zebrafish is a fresh water fish species, it was also susceptible to the tested marine pathogens and their LPS…”  was added to the discussion in Lanes 425-427.

Reviewer 3 Report

The authors of the paper "LPS-induced mortality in zebrafish: preliminary characterization of common fish pathogens" carried out a characterization of LPS and live cell mortality of different fish pathogens. Although the experimental design could be considered correct and the work is of great interest, it lacks in some important parts. Apart from the fact that, in my opinion, they should indicate the amounts loaded in each lane of the SDS-PAGE (although it can be guessed), the fact that they do not observe S-LPS in four strains and that the S-LPS that they observed is different from what they acquired commercially seriously compromises the study. It might be thought that, during its extraction and purification, the O-antigen could be seriously affected (beyond the reasons that the authors discuss, among which is that they used LPS from different strains of P. auroginosa...). Did the authors check if the S-LPS of the four strains could really be affected during the process where it is not seen using any other technique? Why didn't they try to detect it using other techniques such as Western blot or silver staining?

On the other hand, the authors compare mortality results between live cells and LPS. However, it is not surprising that the results do not correlate, since with living cells many other factors may be acting, so I would not include it as a conclusion (since it is a well-known fact). Did the authors study any immunological response in the fish due to the presence of the living cells of the pathogens or due to their LPS?

I would recommend the authors to dispense with the LPS part, until they can demonstrate the presence of S-LPS, and delve a little deeper into the immune response mechanisms presented by fish.

Author Response

We would also like to express our sincere thanks to the reviewer for the constructive comments to improve our manuscript overall quality.

We agree that for clarity reasons, the amounts of LPS loaded in the gel of Figure 1 should also be presented in the figure adding to the description in the Results section, and thus we have added that detail to Figure 1 to improve its understanding.

However, we do not agree that the lack of S-LPS in some of the pathogens analyzed is a sign that our study is compromised. We show that the LPS-extraction methodology used in our study is able to extract both S-LPS and R-LPS, by extracting LPS from a strain of P. aeruginosa, and obtaining a similar profile to P. aeruginosa LPS commercially available. We acknowledge that a different P. aeruginosa strain was used for LPS isolation, and that the profile was not exactly the same as that of the commercial LPS, but the method does clearly isolate both S-LPS and R-LPS. The differences observed in the S-LPS of extracted P. aeruginosa versus commercial LPS, may indeed be related to the use of different strains of the bacterium, as large differences in LPS profile of different P. aeruginosa strains has been previously described (for example in Fig.1 of the following article: http://cmdr.ubc.ca/bobh/wp-content/uploads/2017/02/87.-Rivera-1988.pdf). Additionally, we have previously applied the exact same methodology to extract LPS from different strains of Synechocystis sp. (https://doi.org/10.1111/1751-7915.14057), again obtaining both S-LPS and R-LPS, but with an enrichment of S-LPS (contrary to the profile obtained for the aquaculture pathogens). For all these reasons, it is our belief that extraction issues do not explain the LPS profile obtained for the aquaculture pathogens under study. As said in our discussion, the obtained low-molecular weight R-LPS might in fact represent the real LPS pattern for the strains under study, since the absence of S-LPS is not an unprecedented observation. We give examples of V. harveyi and T. maritimum strains that present only low molecular-weight R-LPS, and although this is the first analysis of P. damselae subsp. piscicida LPS, a closely related species named P. phosphoreum was also reported to present only low-molecular weight R-LPS.

We fully agree with the reviewer that further studies are needed to deeply understand possible LPS contributions to aquaculture pathogens virulence, in particular that of T. maritimum and V. harveyi, whose contribution to pathogens virulence and disease development in fish are unknown. As highlighted in the manuscript title, discussion and conclusion, our study represents an initial characterization of LPS-induced mortality in zebrafish, and our results suggest that aquaculture pathogens LPS contribution to fish diseases is worth investigating. New studies are now underway, targeting the immune response of fish when exposed to these particular LPS.

Round 2

Reviewer 2 Report

The revised Ms can be accepted for being published. 

The English is OK.

Author Response

We thank the reviewer for all the suggestions to improve our manuscript and for accepting it for publication.

Reviewer 3 Report

Although the authors develop the possible reasons why S-LPS is not obtained, they have obviated other considerations made previously (the reasons why they have not tried to detect S-LPS by other techniques and why to use a comparison with living cells). Also, regardless of the possible reasons why they have not obtained S-LPS (or have not detected it), the fact is that internal control of the work has been lost and that any comparison with LPS obtained commercially, from other strain, has no sense. As I said, the work would be of greater interest if, 1) they try to detect S-LPS by other techniques (not just leave with a negative result), and 2) if S-LPS is not detected, the authors could focus only in mortality tests of living strains. Basing the results and discussion on something that may not be seen (a negative result) does not seem appropriate to me.
